# Evaluating Transmission Heterogeneity and Super-Spreading Event of COVID-19 in a Metropolis of China

**DOI:** 10.3390/ijerph17103705

**Published:** 2020-05-24

**Authors:** Yunjun Zhang, Yuying Li, Lu Wang, Mingyuan Li, Xiaohua Zhou

**Affiliations:** 1Department of Biostatistics, School of Public Health, Peking University, Xueyuan Road, Beijing 100191, China; yunjun.zhang@pku.edu.cn (Y.Z.); liyuying@pku.edu.cn (Y.L.); 2Beijing International Center for Mathematical Research, Peking University, Yiheyuan Road, Beijing 100871, China; wangl@bicmr.pku.edu.cn; 3School of Mathematical Sciences, Peking University, Peking University, Yiheyuan Road, Beijing 100871, China; 1700010659@pku.edu.cn; 4Center for Statistical Science, Peking University, Yiheyuan Road, Beijing 100871, China

**Keywords:** COVID-19, super spreading, transmission heterogeneity

## Abstract

COVID-19 caused rapid mass infection worldwide. Understanding its transmission characteristics, including heterogeneity and the emergence of super spreading events (SSEs) where certain individuals infect large numbers of secondary cases, is of vital importance for prediction and intervention of future epidemics. Here, we collected information of all infected cases (135 cases) between 21 January and 26 February 2020 from official public sources in Tianjin, a metropolis of China, and grouped them into 43 transmission chains with the largest chain of 45 cases and the longest chain of four generations. Utilizing a heterogeneous transmission model based on branching process along with a negative binomial offspring distribution, we estimated the reproductive number *R* and the dispersion parameter *k* (lower value indicating higher heterogeneity) to be 0.67 (95% CI: 0.54–0.84) and 0.25 (95% CI: 0.13–0.88), respectively. A super-spreader causing six infections was identified in Tianjin. In addition, our simulation allowing for heterogeneity showed that the outbreak in Tianjin would have caused 165 infections and sustained for 7.56 generations on average if no control measures had been taken by local government since 28 January. Our results highlighted more efforts are needed to verify the transmission heterogeneity of COVID-19 in other populations and its contributing factors.

## 1. Introduction

In December 2019, many cases of viral pneumonia-like disease similar to severe acute respiratory syndrome (SARS) were detected in Wuhan, China. Later, they were confirmed to be caused by a novel coronavirus, provisionally called 2019 novel coronavirus (2019-nCoV). On 30 January 2020, World Health Organization (WHO) declared the 2019-nCoV outbreak as a global health emergency of international concern [1]. On 11 February 2020, WHO named the diseases CoronaVirus Disease 2019 (COVID-19) and announced “severe acute respiratory syndrome coronavirus 2 (SARS-CoV-2)” as the name of the new virus. On 11 March 2020, WHO declared COVID-19 a pandemic. As of 18 April 2020, more than 2,160,207 cases were confirmed globally and the National Health Commission (NHC) of China reported a total of 82,735 cases of COVID-19 in mainland China, including 77,062 recoveries and 4632 deaths.

The dynamics of an infectious disease outbreak depends on both the potential and the heterogeneity of disease transmission. The transmission potential of an infectious pathogen is usually represented by its reproduction number (denoted as *R*) which is the average number of secondary cases caused by a typical infectious individual. Estimates of R>1 (i.e., supercritical outbreak), such as for the early outbreak of the severe acute respiratory syndrome coronavirus (SARS-CoV) in China 2003 [2], indicate the great risk for an infection pathogen to generate a major outbreak; Estimates of R<1 (i.e., subcritical outbreak), such as for the Middle East respiratory syndrome coronavirus (MERS-CoV) in South Korea 2018 [3], imply that the outbreak is slowing down with declining trend of incidence. Though different in transmission potential, both SARS and MERS shared the same feature of high level of heterogeneity which involved the uneven transmission patterns with a number of super-spreading events (SSEs), where some individuals spread to a disproportionate number of individuals, compared to most individuals who infected a few or none [2,4]. SSEs of SARS and MERS were responsible for triggering the initial outbreaks in several large cities such as Beijing, Hongkong, and Singapore, and therefore sustaining the spread of disease worldwide. Besides, SSEs were also documented in many other infectious disease [5]. Detection of transmission heterogeneity may direct prevention efforts and reduce future infections [3,6,7].

The transmission potential of COVID-19 has been studied based on mathematical models, yielding consistent evidence for the high level of reproduction number *R* (1.95–2.2) within a completely susceptible population [8,9,10]. However, evidence for the transmission heterogeneity of COVID-19 and the related SSEs was limited and conflicting. So far, two studies found no heterogeneity in Singapore [11] and in a southern city of China [12], but one study confirmed heterogeneity based on data from 46 countries [13]. In addition, several possible SSEs have been reported in China [14,15], which involved large numbers of infections. Therefore, it would be essential to further explore the transmission heterogeneity, SSEs, and relevant factors of COVID-19.

In this study, we investigated the transmission characteristics of COVID-19 using the epidemiological data of all the confirmed cases in a metropolis in Northern China. We analyzed the transmission chains and then estimated the reproduction number and the transmission heterogeneity using a branching model along with a negative binomial offspring distribution [16]. In addition, we identified the SSE according to the 99-percentile criterion in [2] and assessed the effect of control measures employed by local government.

## 2. Materials and Methods

### 2.1. Data Collection

Tianjin is a municipality and a coastal metropolis in Northern China, with a population of 15.66 million and an area of 11,966 square kilometers. Since the first case of COVID-19 in Tianjin was confirmed on 21 January, a total of 135 cases were confirmed by the time of data collection (26 February 2020) of the current study. From 6:00 on 22 Februaryto 18:00 on 26 February, the city had no new confirmed cases for 108 consecutive hours. The municipal government publicized the epidemiological information of the cases every day. Diagnosis of COVID-19 were based on the protocol issued by China CDC [17].

From the official websites of the Municipal Health Commissions [18], we retrieved data of the 135 confirmed COVID-19 cases in Tianjin, including demographic characteristics and epidemiological characteristics, i.e., travel history and contact history with confirmed/suspicious cases. According to this information, the infection relationship within the scope of Tianjin can be specified. Each case was given a unique number according to its sequential order as reported.

We adopted the definition in [16] to define a transmission chain as a group of cases connected by an unbroken series of local transmission events. We grouped all the confirmed cases in Tianjin into transmission chains and for each transmission chain, we identified the primary case, calculated the chain size (i.e., the total number cases including the primary case), and reconstructed the transmission history (i.e., who infected whom). According to the extent of resolution of the reconstructed information, three types of transmission chains were further identified. The first type was the *simple transmission chain* of which the transmission history could be clearly recovered. The second type was the *ordinary transmission chain* of which the transmission history was not clear but the single primary case could be clearly identified and the chain size was also clear. The third type was the *complex transmission chain* of which the chain size was clear but the primary case could not be clearly identified. More than one individual in the chain exhibited similar behavior/clinical characteristics, so we have to regard them as the primary cases.

As COVID-19 was first reported in Wuhan, a city in the middle of China, a primary case was defined if the individual had a history of travel to or residence in Wuhan within one month, had direct contact with an individual who had confirmed infection outside Tianjin, having fever outside Tianjin, or had the earliest onset of symptoms in the transmission chain.

### 2.2. Analytical Approach

#### 2.2.1. Inference of Transmission Characteristics

To quantify the transmission potential and heterogeneity for the COVID-19 outbreak in Tianjin, we adopted a stochastic model based on branching process to characterize both the distribution of secondary cases (i.e., a negative binomial distribution) and the resulting distribution of transmission chain sizes under the same parameterization of reproduction number *R* and dispersion parameter *k* (lower value indicating higher heterogeneity) for subcritical epidemic [16].

We first fitted the stochastic model to the retrieved information of the simple/ordinary transmission chains. Then, in handling a complex transmission chain with two primary cases, we regarded it could be separated into two ordinary transmission chains, each of which was led by a primary case. The difficulty lay in that the exact size of each ordinary chain was unclear. We dealt with this ambiguity in two ways: one was the combinatorial method in [16] by allowing for all the possible combinations and treated the sum as an overall probability (Further detail can be found in Appendix A.); the other was to adopt the expectation-maximization (EM) algorithm by treating the sizes of two separated chains as latent variables. The EM algorithm also estimated the latent chain size, which was informative for dividing the complex transmission chain into constituent chains (Further detail can be found in Appendix A.).

To identify possible SSEs in the Tianjin outbreak, we adopted the definition in [2] to define a super-spreader as any infected individual causing more infections than would occur in 99% of infectious histories in a homogeneous population. The transmission in a homogeneous situation was modelled with a *Poisson* distribution which is the special case of negative binomial distribution without heterogeneity.

#### 2.2.2. Assessment of Control Measures

Based on the stochastic model accounting for transmission heterogeneity, we assessed the control measures imposed by the Tianjin government. We compared the transmission characteristics (i.e., *R* and *k*) for the periods of before and after the control measures taking effect. Then, based on the estimate of transmission characteristics before control, we simulated the expected distribution of outbreaks that might occur in Tianjin if no control measures were taken. Each simulated outbreak was initiated with 43 infectious individuals (as the number of transmission chains in Tianjin), and was propagated with the branching process model on the basis of inferred characteristics.

## 3. Results

### 3.1. Characteristics of COVID-19 Cases

Among the 135 patients included in the study, 72 (53.3%) were male and 63 (46.7%) female, with an average (standard deviation) age of 47.8 (18.3) years and 50.1 (15.2) years, respectively. As shown by Figure 1A, of the total 135 cases, 34 (25.2%) cases were imported cases. Among the 101 cases of local transmission, the majority (55, 40.7%) were infected in household, 35 (25.9%) cases were infected in public places including a shopping mall and working places, and 11 (8.1%) were unclear in the source of infection. When we examined the chronological development of the infection in Tianjin, we found that the confirmed cases were mainly composed of the imported cases and the household infections in the early and the later stage of outbreak, respectively. The transition happened around 3 February.

### 3.2. Reconstructed Transmission Chains

The 135 cases were grouped into 43 transmission chains including 36 simple chains (47 cases, average size 1.3), 5 ordinary chains (78 cases, average size 15.6), and 2 complex chains (10 cases, average size 5) (Table 1; Figure 2). Detailed information of three types of transmission chains shown in Appendix A.

### 3.3. Estimation of *R* and *k*

Both the combinational method and the EM algorithm gave lower value of *k* with the corresponding 95% confidence intervals (CIs) being lower than 1 (Table 2), suggesting significant evidence for the transmission heterogeneity in the outbreak at Tianjin. In addition, both combinatorial method and EM algorithm gave the same estimate of reproductive number R=0.67 being lower than the critical value of 1, indicating the local transmission would finally die out. Note that these estimates of transmission potential were based on the data by 26 February, implying the average trend over the same period.

When comparing the CIs from different methods, we observed that the EM algorithm generated slightly narrower CIs for both parameters than the combinatorial method. In addition, the EM algorithm had advantage on the ability of providing more information on the split of complex transmission chain, as it also produced a probabilistic insight of unknown partitions of the chain. Recall that the COVID-19 data in Tianjin contains two complex transmission chains of size 5 for each. For these two complex chains, the EM algorithm tended to split it into one size-1 and one size-4 transmission chain (posterior probability = 95.5%), instead of two chains with size 2 and size 3 (posterior probability = 4.5%).

Furthermore, the estimates along with the 95% CIs and confidence regions of combinatorial method were plotted in Figure 3A. The confidence region incorporated the uncertainty in both parameters simultaneously, thus it undoubtedly gave much wider range than the CI, especially for *k*. We also found (Figure 3B) that the observed chain size distribution was close to the expected distribution based on the estimated characteristics (i.e., $\hat{R}=0.67$ and $\hat{k}=0.26$). Particularly, the probability of a chain with size over 5 was less than 10%, and the probability of a single infected case resulting in a chain of size 45 (the largest size of transmission chain in Tianjin) was about 0.8%. It indicated that the outbreak of a comparatively large chain was not likely to happen in Tianjin.

### 3.4. Super Spreading Event in Tianjin

We found notable transmission heterogeneity (k=0.25) in the local spreading of COVID-19 in Tianjin, suggesting that there were likely to be SSEs [5]. As SSEs were more likely to occur at the early stage of the outbreak to trigger out the local spreading in the epidemic of SARS and MERS, we suspect the existence of SSEs at the early stage of outbreak in Tianjin.

Adopting the criterion in [2] to define SSE, we calculated the cut-off as the 99th percentile of the *Poisson* distribution with mean value of 2.2 (i.e., the reproduction number at the early stage). The cut-off value was 6, suggesting that an infected individuals who infected 6 or more secondary cases could be regarded as a super-spreader of the COVID-19 outbreak in Tianjin. According to this criterion, a 57-year-old man, case 2, was a super-spreader. He infected six colleagues in close contact with him before being confirmed on 21 January 2020 (Figure 2). These six colleagues, as secondary cases in this transmission chain, successively infected other colleagues or their relatives.

### 3.5. Effect of Government Control Measures

Tianjin municipal government deployed a series of policies to control the spread of COVID-19 from 28 January 2020, including strong traffic restriction and quarantining individuals who had suspect contacts with confirmed cases [19]. By allowing for the median incubation period of 5.1 days [20], we assumed that the control measures had been taking effect since 1 February 2020.

When we compared the transmission characteristics before and after the intervention taking effect (Table 3), we found that the reproduction number *R* decreased from 0.74 to 0.53, while *k* increased from 0.14 to 0.77, suggesting a decrease in both transmission potential and heterogeneity after taking control measures. However, the overlapping CIs were likely to be caused by the small sample size.

Compared with the observation of 135 confirmed cases in total and the longest transmission chain with 4 generations (Figure 2), the simulation study showed that the local outbreak in Tianjin would have sustained for 7.56 generations and would have led to 165 infections on average if there were no control policies (Figure 4).

## 4. Discussion

Based on individual-level information of COVID-19 infection cases in Tianjin, we discovered significant transmission heterogeneity (k=0.25, 95% CI: 0.13∼0.88) and subcritical transmission potential (R=0.67, 95% CI: 0.54∼0.84), and we identified one super-spreader who infected 6 individuals, suggesting that the local outbreak in Tianjin was considerably heterogeneous even though it would finally die out. In addition, our numerical results successfully verify the effectiveness of government control deployed on 28 January in Tianjin.

Our finding of significant transmission heterogeneity of COVID-19 outbreak in Tianjin was inconsistent with the previous studies conducted in ShenZhen, China (k=0.58, 95% CI: 0.35∼1.18), [12] and in Singapore (k=0.4, 95% CI: 0.1∼Inf) [11]. The absence of heterogeneity in these two studies could have been due to the failure of tracking some epidemiological links among cases. In addition, our result was consistent with another study conducted among 46 countries (k=0.1, 95% CI: 0.05∼0.2) [13]. However, it should be noted that the large-scale spread of COVID-19 is likely to be heterogeneous because many extrinsic factors, such as weather and control measures, may affect the transmission of pathogen. The transmission heterogeneity revealed in the current study was in a local outbreak, which indicates that some intrinsic properties of the pathogen might have contributed to the heterogeneity. This finding is more meaningful for the development of targeted control measures.

Besides the super-spreader in Tianjin, as discussed in the results, some other SSEs were also identified based on our criterion in other cities of China, such as in Ningbo, Zhejiang [21]; in Harbin, Heilongjiang [22]; and in Hong Kong [23]. With the tendency of pandemic of COVID-19, all governments should strengthen controls to prevent SSEs.

Previous studies identified multiple underlying reasons for the emergence of SSEs and high transmission heterogeneity [5]. For example, SSEs of MERS-CoV and SARS-CoV were largely driven by the environmental and clinical factors, such as hospital transfer, substantial delay of diagnosis, and fundamental crowding of population [4,5]. The transmission heterogeneity and the identified SSE of COVID-19 in Tianjin, however, exhibited different features. The SSE happened in a working place, and there was no obvious delay of diagnosing the super-spreader (3 days from onset of symptom to diagnosis). In addition, the longest transmission chain (as shown in Figure 2) occured after the initiation of level one response of public health emergency by the Tianjin government. Moreover, the public were advised to avoid mass gathering and stay at home. Obviously, the aforementioned reasons for the transmission heterogeneity in MERS-CoV and SARS-CoV did not hold in the COVID-19 outbreak. Therefore, we suspected the transmission heterogeneity of COVID-19 were driven by some other epidemiological characteristics, such as asymptomatic transmission and contagiousness during incubation period, i.e., transmission can occur before symptom onset. The onset of viral shedding prior to the onset of symptoms, or in cases that remain asymptomatic, is a classic factor that makes infectious disease outbreaks difficult to control [24]. A recent reported transmission chain of large size in Harbin, China, was triggered by asymptomatic infections [25], which justified our speculation and highlighted the need for efficient measures such as rapid testing suspected cases to reduce the transmission from subclinical cases.

Additionally, our estimate of transmission potential (R<1) suggested that the spreading of COVID-19 would not cause large infection in Tianjin. This estimate was the average trend over the period of data collection (from 21 January to 26 February), and was considerable lower than that of other studies, especially at the early stage of COVID-19 outbreak [10]. The differences in the reproductive number reported from different studies are largely due to differences methods, differences in data sources and time periods used to estimate the reproductive number. In addition, as indicated with our evaluation based on simulation, the value of R<1 indicated that the control measures imposed in succession by the local government had been considerable to reduce the spread of virus.

Regarding the effect of control measure undertaking by the local government, there were changes in point estimates of *R* and *k* of before and after government control, but the difference was not statistically significant with overlapping confidence intervals. Even though, by considering the resulted decrease of 19% infection (from 165 cases to 135 cases) within 25 days, these control measures still had some practical implication.

Although the spread of COVID-19 is currently under control in China, it is still facing the risk and challenge brought by the resumption of work and imported cases from other countries. The verification of transmission heterogeneity and SSE in Tianjin outbreak of COVID-19 reminds the government to pay much attention to asymptomatic transmission and other factors that may lead to the transmission heterogeneity and SSEs. Furthermore, more efforts are also needed to explore the emergence of SSE and its contributing factors.

It is also worth mentioning the limitations of our study. Totally 34 imported cases were reported in Tianjin during the period of our study, but we identified 43 transmission chains, which meant some epidemiology links among cases were missing. The estimate of heterogeneity might be driven up if the confirmed cases were condensed into fewer transmission chains. Meanwhile, we only analyzed COVID-19 data from Tianjin (135 confirmed cases), a relatively small sample size compared with the total number in China (exceeds 80,000 confirmed cases), and our case data might be subject to bias or under-reporting. However, this could not alter our final conclusion of significant heterogeneity, since a study [26] explored small-sample bias and under-reporting bias and found that maximum likelihood estimates of k can be biased upward but are not biased downward.

## 5. Conclusions

In conclusion, we proved that the transmission of COVID-19 is heterogeneous and identified the existence of one SSE in Tianjin. We also showed that the control measures undertaken by the local government effectively alleviated the outbreak in terms of infection size and duration. As a pandemic which is still spreading worldwide at a startling speed, the transmission characteristics of COVID-19 needs more exploration and investigation in a large scale.

## Figures and Tables

**Figure 1 ijerph-17-03705-f001:**
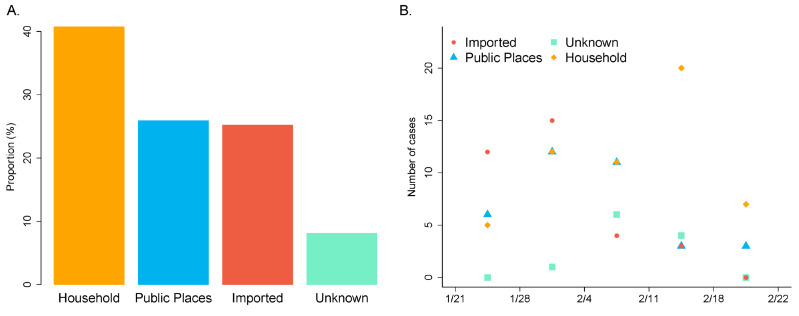
Case information in Tianjin (totally 135 cases, reported from 21 January to 26 February). (**A**) Proportion of cases infected in different ways/places. (**B**) Chronological development of the infection by transmission ways/places.

**Figure 2 ijerph-17-03705-f002:**
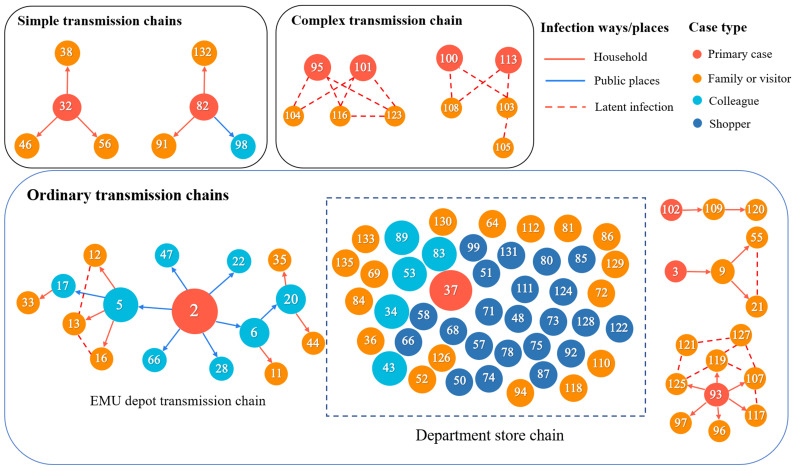
Reconstructed transmission chains (excepted for isolated cases) of COVID-19 outbreak in Tianjin by 26 February 2020. The red circles represent the primary cases in each chain, the orange circles are the family members or visitors, and the blue circles are the colleagues. The red arrows and the blue arrows represent the transmissions within household and in public places, respectively. The dash lines represent latent epidemiological links. Besides, the dotted box indicates that all cases in the department store chain have the potential to infect others.

**Figure 3 ijerph-17-03705-f003:**
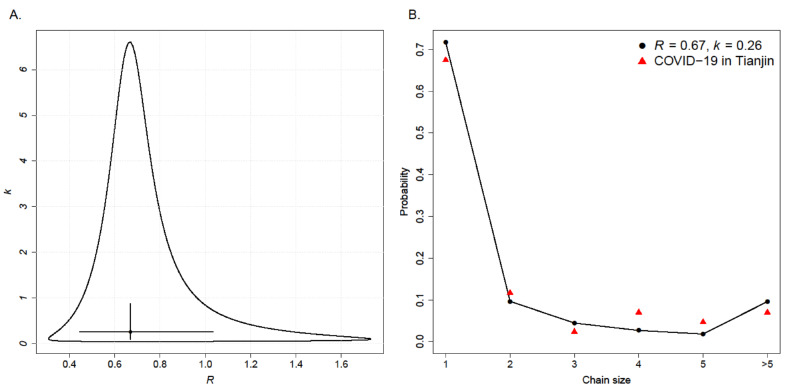
Analysis of COVID-19 outbreak in Tianjin using the combinatorial method. (**A**) The circle, cross hair, and curve represent the estimates, 95% confidence intervals, and confidence region of parameters *R* and *k*, respectively. (**B**) Circles denote the probability of a transmission chain with size from 1 to >5 based on the estimates of *R* and *k*; Triangle denotes the frequency of a transmission chain with corresponding size in Tianjin COVID-19 data.

**Figure 4 ijerph-17-03705-f004:**
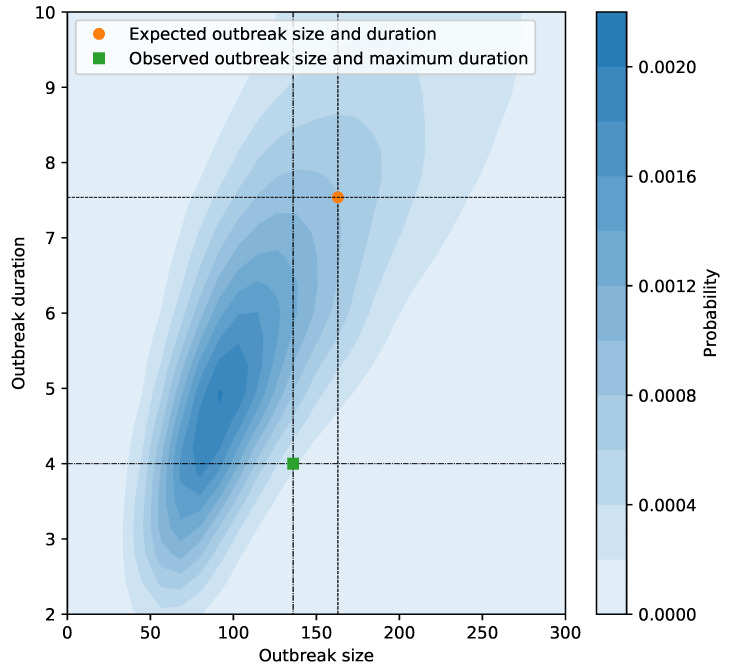
Simulated outbreak size and duration by assuming no control measures. Each simulation was started with 43 infections and based on reproductive number R=0.74 and dispersion parameter k=0.14, which were estimated from the data collected by 1 February 2020. The density and mean of duration and outbreak size were estimated based on 5000 Monte Carlo simulations.

**Table 1 ijerph-17-03705-t001:** Three types of COVID-19 transmission chains in Tianjin.

Chain Type	Amount	Total Number	Average	Range of
	of Chains	of Cases	Chain Size	Chain Size
Simple transmission chain	36	47	1.3	1–4
Ordinary transmission chain	5	78	15.6	3–45
Complex transmission chain	2	10	5	5–5

**Table 2 ijerph-17-03705-t002:** Estimation and CI of the reproduction number *R* and dispersion parameter *k* based on the conbinational method and the expectation-maximization (EM) algorithm.

	Combinatorial Method (95% CI)	EM (95% CI)
*R*	0.67 (0.44, 1.03)	0.67 (0.54, 0.84)
*k*	0.26 (0.10, 0.88)	0.25 (0.13, 0.88)

**Table 3 ijerph-17-03705-t003:** Estimation of the reproductive number *R* and the dispersion parameter *k* for different periods.

	Before 1 February (95% CI)	After 1 February (95% CI)
*R*	0.74 (0.39, 1.61)	0.53 (0.29, 0.96)
*k*	0.14 (0.04, 0.63)	0.77 (0.14, 31.47)

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
