# Peer review of "Evaluating Transmission Heterogeneity and Super-Spreading Event of COVID-19 in a Metropolis of China"

_ijerph, 2020, doi:10.3390/ijerph17103705_

Round 1

Reviewer 1 Report

The authors acquired public data that showed 135 cases of Covid-19 from Jan. 21 to Feb. 26, 2020 (5 weeks or 37 days) and examined this data to determine the transmission characteristics of Covid-19 in Tianjin, China. The authors traced these infections to 43 transmission chains and further subdivided these chains to provide evidence heterogeneity of pathogen spread and the presence of at least one super spreader event. The manuscript is well-written, the statistical methods are sound, and data are properly interpreted, albeit, the sample size is small. However, the data is relevant and useful to support intervention methods are reducing the reproductive number (r) and the dispersion factor (k) (heterogeneity). Hence, this report is a valuable contribution to better understand of the epidemiology Covid-19 and the importance of social distancing for its control.

P5. Fig. 2. Suggest adding letters to differentiate different figures in the group. For example, A for Simple; B for Ordinary, C for Complex; D for EMU and E for Dept store.

Ln 161-164. Even though the Poisson distribution cutoff showed only one SSE, Fig. 2 EMU depot appears to show 2, one that resulted in 116 infections and the other 96 infections. Clarify.

Ln 170. Correct typo for strong.

Lns 194-197. Awkward sentence. Rewrite.

Ln 227. Add “to” after considerable and replace “the transmissibility” with “spread”

Ln 247 …which could not destroy….??? Yes, the small sample size can significantly affect heterogeneity. Rephrase this statement.

Reviewer 2 Report

This manuscript describes use of a stochastic model to calculate the basic reproduction number of SARS-CoV-2 spread in Tianjin, China. The authors find substantial heterogeneity in the transmission chains of Tianjin and find a basic reproduction number of less than 1. The study is interesting, relevant, and done well. I have two major suggestions for improvement before the manuscript can be published:

  1. The manuscript needs some editing of the English. There are enough grammar problems that it detracts from clear presentation of the results.
  2.  The authors need to provide a more detailed description of the stochastic model they are using (equations would be good here) including clearly indicating how they are getting R_0 and k from the model.
